# Effects of Self-Care Education Intervention Program (SCEIP) on Activation Level, Psychological Distress, and Treatment-Related Information

**DOI:** 10.3390/healthcare10081572

**Published:** 2022-08-18

**Authors:** Nor Aziyan Yahaya, Khatijah Lim Abdullah, Vimala Ramoo, Nor Zuraida Zainal, Li Ping Wong, Mahmoud Danaee

**Affiliations:** 1Department of Nursing Science, Faculty of Medicine, Universiti Malaya, Kuala Lumpur 50603, Malaysia; 2Department of Nursing, School of Medical and Life Sciences, Sunway University, Petaling Jaya 47500, Malaysia; 3University Malaya Specialist Centre, Kuala Lumpur 59100, Malaysia; 4Department of Social and Preventive Medicine, Faculty of Medicine, Universiti Malaya, Kuala Lumpur 50603, Malaysia

**Keywords:** self-care education, physical and psychological symptoms, activation level, psychological distress, treatment-related information, female breast cancer, symptom management, longitudinal study

## Abstract

Self-care education can direct patients to manage their side effects during treatment, reduce psychological distress, and improve self-care information. In this study, the effectiveness of the Self-Care Education Intervention Program (SCEIP) on patient activation levels, psychological distress, and treatment-related concerns in women with breast cancer was assessed by adopting a longitudinal quasi-experimental pre-test and post-test design. The data for 246 women with breast cancer undergoing adjuvant chemotherapy were collected. Pre- and post-interventional assessments were conducted at baseline (T1) and the second (T2), fourth (T3), and sixth (T4) cycles using the 13-item Patient Activation Measure, 14-item Hospital Anxiety and Depression Scale, and 25-item Cancer Treatment Survey. It was found that the SCEIP significantly improved the activation level (*p* ≤ 0.001), psychological distress (anxiety level (*p* ≤ 0.001), the depression level (*p* ≤ 0.001)), and treatment-related concerns (sensory/psychological concerns (*p* = 0.05); procedural concerns (*p* ≤ 0.001)). Therefore, the SCEIP could potentially improve patients’ activation level, psychological distress, and treatment-related concerns regarding symptom management during chemotherapy, specifically for Malaysian women with breast cancer.

## 1. Introduction

Breast cancer is one of the most common cancers worldwide that affects post-pubescent women. Breast cancer treatments, such as chemotherapy, may cause undesirable side effects for patients, including nausea and vomiting, dry mouth, taste changes, loss of appetite, weight changes, diarrhea, alopecia, infection, and energy loss. These side effects can influence patients’ self-care abilities, causing them to neglect their physiological and psychological welfare [1,2]. Therefore, self-care during cancer therapy is an essential non-pharmacological approach to improving the well-being of patients living with breast cancer [2,3,4]. Furthermore, self-care includes activities that individuals initiate and engage in to maintain their quality of life, health, and well-being [4,5]. Self-care management practices include opting for a healthy lifestyle, self-monitoring, assessing symptoms, evaluating symptom severity, and determining treatment alternatives [4]. The literature on self-care management in Western countries has consistently found that patient engagement in various self-management strategies is vital for a successful symptom management program [6,7,8]. Moreover, patient involvement in self-management was found to be beneficial in alleviating symptoms of depression, anxiety, and emotional distress [7,8,9,10,11,12]. Therefore, nurses have a significant role in empowering patients to engage, manage, and improve their health through self-care management practices, particularly by enhancing patient activation. 

Patient activation emphasizes patients’ willingness and ability to take independent actions to manage their health and care [13]. In other words, patient activation relates to their commitment to preventive and healthy behaviors and to treatments. This step also refers to the individuals’ knowledge skills and behavioral repertoire to manage their condition by collaborating with their healthcare providers to maintain their health function and access appropriate and quality care [14,15]. Previous empirical studies indicated that active patients were significantly more likely to attend screenings, regular check-ups, treatments, and immunizations and engage in healthy behaviors such as maintaining a healthy diet [14,15,16,17] or regular exercise [18,19,20,21,22] than those who scored lower on the activation scale.

Patient activation strategies include improving patients’ knowledge, confidence, and/or self-management skills [13]. Providing information to patients is crucial in engaging them in self-care management and constitutes essential supportive care across the cancer continuum. The goal of providing the information is to prepare patients for treatment, increase treatment adherence, improve the ability to cope with disease, and promote recovery [13,14,15]. Educating patients about their treatment reduces anxiety, increases self-confidence, improves compliance, and increases their participation in self-care [5,10]. In addition, a meta-analysis of randomized controlled trials found that providing sensory and procedural information lowered anxiety, pain, and distress levels and promoted a rapid recovery with fewer complications [23]. The review also emphasized that sensory and procedural information were two critical components required to address pre-treatment anxiety or fears, particularly in painful medical or surgical procedures. 

Procedural information refers to the details of the hospital environment, such as the location; the chain of events; and precautionary measures taken before, during, and after the procedure. Meanwhile, sensory information refers to what the patient is likely to experience before, during, and after the procedure, such as sensations including pressure, type of pain, and sounds of machinery [24]. Combined procedural and sensory preparations yield the most robust patient outcomes [23] for invasive procedures [25,26] such as chemotherapy and radiotherapy [27,28]. Patients must be prepared for cancer treatments to deal with fears and misconceptions about the procedure. In addition, they must understand and anticipate normal sensations during and after treatment and cope with post-treatment side effects. Therefore, the two information components should be included in educational strategies for self-care management.

Numerous Western studies have developed effective evidence-based self-care management interventions, but only a few studies have been conducted on the Asian population. Perceptions of symptom experiences and self-care strategies in managing symptoms and outcomes may vary for patients in a multiracial country such as Malaysia, which is geographically and culturally different from its Western counterparts. Therefore, this study aimed to assess the effectiveness of the Self-Care Education Intervention Program (SCEIP) on patient activation levels, psychological distress, and treatment-related concerns in women with breast cancer. This study was designed based on the Revised Symptom Management Model by Dodd et al. [29], which is highly comprehensive in dealing with patients’ symptoms. The research model is based on cancer patients’ descriptions of their symptom experiences, development of symptom-management strategies, and determining their effectiveness. We hypothesized that the SCEIP would equip female breast cancer patients with the essential knowledge and skills required to achieve optimal symptom management during treatment, improve activation levels and treatment-related concerns, and alleviate psychological distress during chemotherapy. 

## 2. Materials and Methods

### 2.1. Study Design, Participants, and Procedures

A quasi-experimental longitudinal pre-test and post-test were conducted on women with breast cancer undergoing adjuvant chemotherapy at a regional cancer center in Putrajaya, Malaysia. This study was conducted in two stages: Stage 1 included procedures for the control study, and Stage 2 included the intervention procedures. This study design and the two-stage approach were chosen instead of a randomized controlled study due to several methodological challenges [30,31]. First, the patients from both groups attended the same day care center; hence, the risk of information contamination could dilute the intervention effect [32]. Secondly, the nurses for the study intervention could not be blinded due to the limited staff in charge of caring for patients in the study setting [33,34]. The sample size for this study was 246 participants (α = 0.05, Power (1-β err probe) = 0.95) as calculated using the G-Power software version 3.1 for repeated-measures ANOVA (within-between interactions). This study used the thresholds proposed by Cohen [35] to interpret the effect size. 

The breast cancer patients at the day care center during the study period were recruited based on the following inclusion criteria: (i) subjects diagnosed with breast cancer and undergoing a standard adjuvant chemotherapy regimen for the first time (standard and high-risk chemotherapy regimen: FEC (5-fluorouracil, epirubicin, cyclophosphamide)/FAC (5-fluorouracil, adriamycin, cyclophosphamide) taxane-based or not for six cycles lasting 21 days each with a rest period between each cycle = a total of 18–20 weeks); (ii) 18 years or older; (iii) absence of cognitive impairment; (iv) Eastern Cooperative Oncology Group (ECOG) performance status at grade 1–2; (v) able to speak and write in the Malay or English language; and (vi) no medication or any procedure restrictions during treatment. Participants were excluded if they had prior chemotherapy experience, started chemotherapy concurrently with radiotherapy, needed assistance with their daily activities and treatment therapy, had breast lymphedema, received coaching by any nurse navigators in the study setting, or had been treated with experimental therapy in a clinical trial. 

The data collection took place in two stages within 16 months. A list of eligible patients for the study was provided to the researchers in both stages by the nursing staff (Figure 1). Patient recruitment for the control group (CG) occurred in Stage 1 between June 2017 and early January 2018, while patient recruitment for the intervention group (IG) in Stage 2 took place between early February 2018 and late September 2018. The recruitment process continued until a maximum sample size was reached for both groups. A one-month break was allowed between stages to ensure that no overlapping or residual samples could contaminate the study results. The questionnaires were distributed to the patients at four time points (baseline (T1), second cycle (T2), fourth cycle (T3), and sixth cycle (T4)) for both stages by the nursing staff in charge of the day care center. Patient information sheets were given and explained to both groups, but the intervention process was only revealed to the IG. The primary researcher conducted all study-related information deliveries, interviews, and follow-up discussions in the counseling room before chemotherapy sessions to avoid interruptions. Reporting of this study was per the Statement on Consolidated Standards of Reporting Trials (CONSORT).

A total of six patients dropped out of the study because their treatment had been switched to palliative care (CG, *n* = 3 and IG, *n* = 3). Thus, the total number of patients at the end of treatment was 246 (CG, *n* = 123 and IG, *n* = 123), resulting in a response rate of 97.6%.

### 2.2. Intervention

The SCEIP was designed based on a Phase 1 qualitative study [36] to alleviate psychological distress, increase patient activation, and evaluate treatment-related concerns during chemotherapy. In addition, the SCEIP was developed using the intervention-mapping approach [37,38], which emphasizes developing an intervention program that illustrates the relationship between actions and patient outcomes using suggested elements such as patient problems and intervention measures. Furthermore, the SCEIP intervention is tailored to the individual’s needs and focuses on four self-management goals: addressing physical and psychological symptoms, performing self-care strategies, monitoring symptoms and recording symptoms promptly, and reporting and discussing symptoms with healthcare providers. Motivational interviewing (MI) was used as a coaching strategy during the face-to-face educational session, during WhatsApp chats, and throughout the treatment, especially when meeting with the patients to achieve the intervention objectives. The MI originated from the Transtheoretical Model, which posits that people are at different stages of readiness to make behavioral changes [12,39,40]. 

The SCEIP consisted of two intervention sessions conducted over three weeks according to the patient’s chemotherapy regimen. The first session was divided into two parts: educational and self-management plans conducted via interviews with patients using the basic MI principles regarding their attitude, motivation, and confidence while implementing the self-management strategies. Baseline questionnaires for preliminary assessment of treatment-related concerns, psychological distress, and activation levels were collected before the first session. In the first 30 min, patients received a detailed education on these three aspects: (i) the definition of chemotherapy, the purpose of chemotherapy, how chemotherapy works, the type of chemotherapy regimen they would receive, the chemotherapy duration, the location where chemotherapy would be administered, and the reasons for side effects; (ii) anticipated side effects after chemotherapy based on the literature; and (iii) self-care strategies including diet, lifestyle changes, natural treatments, mind control practices, and pharmacological intervention. Side effects were divided into two main categories: physical and psychological symptoms. Physical symptoms included nausea and vomiting, fatigue, dry mouth, loss of appetite, taste change, stomach bloating/gastritis, constipation, diarrhea, weight loss, hand and foot syndrome, body aches, headache or giddiness, alopecia, skin and nail changes, dry skin, and bone marrow suppression. Meanwhile, psychological symptoms included emotional distress, difficulty sleeping, difficulty concentrating, and agitation. The structured information was presented to the patients while considering their literacy level using a PowerPoint presentation on an Android tablet to improve their understanding.

The second intervention session was implemented at home after patients received chemotherapy until the completion of their treatment. Tools such as a self-care diary, WhatsApp chat group, and personal calls were utilized for these sessions. Topics of discussion included how to use the self-care diary at home, where patients were required to report 20 physical and psychological symptoms over three weeks and rate the symptoms severity and distress. Moreover, the diary contained a list of self-care strategies patients could practice at home. The researcher also explained how the information should be organized in the diary and the deadline for completing and submitting the diary. Additionally, the researcher sent short messages twice a week (on Monday and Thursday) via the WhatsApp chat group and personal chats to ask patients about the symptoms they experienced and how they were coping. Reinforcements of the use of self-care strategies were given via WhatsApp. Patients were also encouraged to talk about their symptom management in the group and whether the self-care strategies were helpful. The time taken for each WhatsApp conversation was between 10 and 15 min. 

The intervention was supplemented with written materials in the form of a patient self-management booklet entitled *Chemotherapy and You: Managing Your Side Effects at Home*. The booklet contained the following information: basic knowledge about chemotherapy, types of chemotherapy, the chemotherapy procedure, 20 physical and psychological symptoms related to the adjuvant breast cancer chemotherapy regimen, and self-care strategies to reduce symptom burden at home. The self-care strategies were related to diet, lifestyle changes, natural treatments, mind control practices, and pharmacological interventions for breast cancer patients. The booklet was printed in Malay and English using simple language to ensure patients could easily understand the content. Both versions were validated by a panel of experts consisting of a clinical oncologist, a nursing lecturer with experience in the field of oncology nursing, a ward and clinical manager working in the oncology clinic, and a breast cancer survivor who had been involved in a Phase 1 qualitative study [36]. The researcher conducted the study interventions, while nurses in charge of patient care assisted in the questionnaire distribution.

### 2.3. Standard Care

Patients in the control group received standard treatment from the oncologists and nurses at the National Cancer Institute (NCI), Putrajaya, Malaysia. Standard practice for cancer patients undergoing chemotherapy at NCI’s Day Care Oncology included providing several cycles of chemotherapy during a span of four months to a year. The chemotherapy regimen was administered according to the different types of cancer. A pre-education session on chemotherapy was held in one of the conference rooms before starting the treatment. Verbal instructions on the treatment plan and follow-up care were given by the nurse in charge of education, while the pharmacists provided information on chemotherapy side effects, medications, and advice on routine healthcare to a group of patients with various types of cancer. 

During each hospital visit, the doctor assessed the patient’s symptom experience at home by asking them questions, and treatments were provided based on their problems or complaints. Both groups received supportive care according to the local practice guidelines and clinical judgement by the doctors and nurses. At the end of the data-collection period, the CG patients received the booklet provided to the IG patients at the beginning of the intervention to ensure that all patients were exposed to side-effect management strategies and practiced self-care at home. Furthermore, the researcher was aware of the potential response burden to the clinical study; thus, efforts were made to balance scientific interests and patient assessment. For instance, questionnaires were completed at an average of less than 35 min at the baseline and follow-up visits. Furthermore, the nurse in charge of both groups was trained to ensure that all patients were fully informed about the time required to complete the questionnaires. Consequently, there were no reports by patients about feeling overwhelmed or distressed in completing the questionnaires as a reason for withdrawal. In addition, the questionnaires were designed in a clear format, and patients could seek assistance from the nurse in charge if required.

### 2.4. Research Instruments

The questionnaire used in this study consisted of four sections. Section one comprised patients’ demographic and medical characteristics (age, ethnic group, educational level, marital status, employment status, income level, menopausal status, ECOG performance status, cancer stages, and chemotherapy regimen). Sections two, three, and four consisted of the Cancer Treatment Survey (CaTS), Hospital Anxiety Depression Scale (HADS), and Patient Activation Measure (PAM), respectively.

#### 2.4.1. Cancer Treatment Survey (CaTS)

The CaTS measured the patients’ treatment-related concerns. This survey, which was developed and tested by Schofield et al. [24], has been widely used and validated worldwide. The questionnaire consists of 25 items with two subscales: 14 items on sensory-psychological concerns (SPC) and 11 items on procedural concerns (PC). Patients indicate the extent to which they agree or disagree with the items using a five-point Likert scale (1—strongly disagree to 5—strongly agree), with higher scores indicating a greater need for assistance [24]. The original version of CaTS was translated from English to Malay with forward and backward translations. The Malay version was then compared and revised to reduce conceptual transcultural content differences [41,42]. The CaTS content validation was conducted by a panel of experts consisting of an oncologist, a clinical nurse specialist, a senior nurse lecturer, and two experienced oncology nurses. No changes were made to the questionnaires after validation.

#### 2.4.2. Hospital Anxiety and Depression Scale (HADS)

Section three consisted of the HADS [43], which assessed the patients’ psychopathological co-morbidity. This questionnaire includes 14 items divided into two subscales for anxiety (HADS-A: 7 items) and depression (HADS-D: 7 items). Each item is rated on a 4-point Likert scale (0 to 3), accumulating a maximum score of 21 for anxiety and depression, respectively. The option for every item varies; the cut-off score for each subscale is ≥11, indicating a “probable case” of anxiety or depression. Nevertheless, a recent systematic review recommended that the best threshold for HADS-D was 5 (sensitivity 0.84, specificity 0.50), and 7 or 8 for HADS-A (sensitivity 0.73, specificity 0.65) in cancer patients [44]. The psychometric properties of HADS (Malay version) are reported based on the internal consistency reliability (Cronbach’s alpha). The internal consistency for the full scale of the Malay HADS questionnaire was 0.87, the anxiety subscale was 0.81, and the depression subscale was 0.73. The overall findings suggested that the HADS demonstrated adequate evidence of reliability [45].

#### 2.4.3. Patient Activation Measure (PAM)

Section four consisted of the PAM, which assessed patient activation or ability to self-manage their health. The 13 items in the PAM form a unidimensional construct of knowledge, skills, and confidence for self-management. A high internal consistency (α = 0.9) and excellent discriminative validity have been reported based on known group comparisons [46]. Each item has four possible response options ranging from 1 (strongly disagree) to 4 (strongly agree) and an additional “not applicable” option. The raw score was divided by the number of items answered (excluding “not applicable” items) and multiplied by 13 to calculate the total PAM score. Based on the calibration tables, this score was transformed into a scale with a theoretical range of 0 to 100; higher PAM scores indicated higher patient activation [46]. The Malay version of the PAM was translated by a group of native Malaysians living in New York, and the reported reliability test result (Cronbach’s alpha) was 0.76 [46]. The reliability of the instruments was tested through a pilot study with 30 samples. The Cronbach’s alpha coefficient of all instruments was ≥0.6, suggesting acceptable internal consistency reliability for the scales [46].

#### 2.4.4. Ethical Consideration

The study was conducted at a regional cancer center in Putrajaya, Malaysia, after obtaining ethical approval from the Institutional Review Board of the cancer center (NMRR-16-1529-30442(IIR)). The patients participated in the study voluntarily, and no rewards nor incentives were offered for their participation. Written informed consent was obtained from the patients after they were briefed about the purpose and methodology of the study. Anonymity and confidentiality of the patient’s data were safeguarded through secure storage. The data will be kept for five years after the study’s completion and then discarded according to hospital policy.

### 2.5. Data Analysis Method

The collected data were analyzed using the Statistical Package for the Social Sciences (SPSS) version 25. All demographic and medical variables of each group were examined separately using descriptive statistics (frequency and percentage). An independent *t*-test and chi-squared test were performed to compare the CG and IG demographic and medical characteristics. Furthermore, the groups were compared using a two-way repeated measure analysis of variance (RM-ANOVA)/repeated measure analysis of covariance (RM-ANCOVA) and multivariate analysis of covariance (RM-MANCOVA), followed by a Bonferroni test for mean comparison at *p* < 0.05. Significant demographic and medical variables in the *t*-test and chi-squared test were used as covariates in the ANCOVA and MANCOVA analysis. All statistical assumptions were tested and met using the normality test, homogeneity test of variance, sphericity test, and homogeneity of regression slopes before the data analyses were conducted.

## 3. Results

### 3.1. Demographic and Medical Variables of Intervention and Control Groups

As shown in Table 1, there were no significant differences between the patients in the two groups for all demographic and medical variables. The homogeneity of the research variables for both groups was also established using an independent sample *t*-test. The results indicated no significant differences between CG and IG for PAM, HADS, and CaTS at the baseline stage.

### 3.2. The Effectiveness of the SCEIP on Study Variables

#### 3.2.1. Patient Activation Measure

This section of the study demonstrated the impact of the SCEIP in improving the activation levels of breast cancer patients undergoing chemotherapy based on the PAM at baseline (T1), cycle 2 (T2), cycle 4 (T3), and cycle 6 (T4). A two-way RM-ANOVA was used because there was no significant relationship between the PAM and the demographic and medical variables. We found that there were significant main effects for group (*F*_(1242)_ = 32.88, *p* ≤ 0.001, partial *ƞ*^2^ = 0.1274), time (*F*_(2.55,618.1)_ = 4.56, *p* ≤ 0.001, partial *ƞ*^2^ = 0.018), and interaction between time and group (*F*_(2.55,618.1)_ = 12.56, *p* ≤ 0.001, partial *ƞ*^2^ = 0.049). Thus, the changes in the PAM among patients of both groups were significantly different across time (baseline, cycle 2, cycle 4, and cycle 6). A pairwise comparison of the total PAM scores at baseline until the cycle 6 follow-ups was also conducted to investigate the actual differences that occurred with a significant level of *p* = 0.005 (two-tailed) after the Bonferroni adjustment (Table 2).

#### 3.2.2. Psychological Distress

This section of the study exhibited the effect of the SCEIP in reducing the psychological distress of breast cancer patients undergoing chemotherapy at baseline (T1), cycle 2 (T2), cycle 4 (T3), and cycle 6 (T4) based on the HADS. Since there were significant relationships between anxiety and depression levels with demographic characteristics (age, marital status, ethnic group, employment status, education level, and income level) and the heterogeneity of groups in terms of income level at the baseline, these variables were considered as covariates in the analysis. Therefore, the RM-MANCOVA was performed to assess whether there were differences between groups and over time in the patients’ psychological distress levels.

There were no significant main effects on group anxiety levels (*F*_(1571.4)_ = 0.45, *p* = 0.50, partial *ƞ*^2^ = 0.002) or time (*F*_(2.41,571.4)_ = 2.35, *p* = 0.09, partial *ƞ*^2^ = 0.01), whereas the interaction between time and group was found to be significant (*F*_(2.41,571.4)_ = 7.08, *p* ≤ 0.001, partial *ƞ*^2^ = 0.029). Thus, the changes in anxiety levels among patients in both groups significantly differed across time (Table 3). Meanwhile, there were no significant main effects for group for depression levels (*F*_(1649.7)_ = 1.70, *p* = 0.19, partial *ƞ*^2^ = 0.007) or time (*F*_(2.74,649.7)_ = 2.10, *p* = 0.11, partial *ƞ*^2^ = 0.009), but the interaction between time and group was significant (*F*_(2.74,649.7)_ = 5.34, *p* ≤ 0.001, partial *ƞ*^2^ = 0.022) (Table 3). Therefore, the changes in patients’ depression levels for both groups significantly differed over time. Pairwise comparison of HADS scores at baseline until the cycle 6 follow-ups were conducted to investigate the actual differences that occurred at a significant level of *p* = 0.005 (two-tailed) after the Bonferroni adjustment (Table 3).

#### 3.2.3. Cancer Treatment Survey

This section of the study showed the impact of the SCEIP in reducing cancer-treatment-related concerns of breast cancer patients undergoing chemotherapy. The analysis was performed using the two-way RM-ANOVA based on the CaTS results at baseline (T1) and at end of treatment (cycle 6 (T4)) to evaluate the effect of teaching and coaching given in pre- and post-chemotherapy. The results showed that for sensory/psychological concerns (CaTS-SPC), there were significant main effects for group (*F*_(1244)_ = 3.84, *p* = 0.05, partial *ƞ*^2^ = 0.016), time (*F*_(1244)_ = 111.03, *p* ≤ 0.001, partial *ƞ*^2^ = 0.313), and interaction between time and group (*F*_(1244)_ = 3.25, *p* = 0.05, partial *ƞ*^2^ = 0.013). Based on the Bonferroni test results, the mean difference scores of CaTS-SPC between T1 and T4 were significantly different for IG (*p* ≤ 0.001) and CG (*p* ≤ 0.001). The difference between groups at T1 (*p* = 0.955) was not statistically significant, but there was a statistically significant difference in the mean score of CaTS-SPC at T4 (*p* = 0.042) (Table 4).

We also found that there were significant main effects between groups in terms of procedural concerns (CaTS-PC), (*F*_(1244)_ = 19.16, *p* ≤ 0.001, partial *ƞ*^2^ = 0.073), time (*F*_(1244)_ = 115.95, *p* ≤ 0.001, partial *ƞ*^2^ = 0.322), and interaction between time and group (*F*_(1244)_ = 27.94, *p* ≤ 0.001, partial *ƞ*^2^ = 0.103). Thus, the changes in procedural concerns (CaTS-PC) among patients in both groups significantly differed over time. Later, the Bonferroni post hoc test was applied to compare the total mean scores; there was a statistically significant within-group difference for IG and CG (*p* ≤ 0.001) in total mean score for CaTS-PC between T1 and T4, while at baseline (T1), there was no significant difference between the intervention and control groups for CaTS-PC (*p* = 0.25), but at T4, there was a statistically significant difference between the groups (*p* ≤ 0.001) (Table 4).

## 4. Discussion

The study findings revealed that the strategies and support provided through the SCEIP effectively increased patient activation levels and reduced psychological distress and cancer-related concerns. A plausible explanation for these positive outcomes could be that the intervention incorporated activities curated for the patients’ actual needs [11,12]. The program allowed patients to connect emotionally and psychologically with their ongoing treatment. In addition, the WhatsApp chat group platform for coaching, self-care diary, and chemotherapy booklet, in addition to the structured education of the SCEIP, could have helped prepare patients and build their confidence in self-managing their symptoms during treatment. Active communication via WhatsApp with healthcare workers during treatment, particularly with the oncology nurse, was essential for patients to avoid distress and lack of confidence in managing symptoms at home and the burden of side effects [7].

Notably, the long-term intervention improved patients’ activation levels in the IG compared to CG. Furthermore, the findings suggested that patients with higher activation scores were more likely to exhibit self-care management behaviors, practice self-care strategies as suggested, and demonstrate compliance in completing the chemotherapy. The current study was designed according to an earlier report that suggested the intervention period should be longer for studies focusing on self-management, ranging from nine weeks to six months [47]. Moreover, encouraging patients to take appropriate small steps toward a successful recovery can enhance behavioral changes. The sense of accomplishment can motivate patients to continue building the skills and confidence needed for self-management [48,49,50].

The literature showed that education-based interventions alone were insufficient to stimulate behavioral change and self-management among patients. In contrast, vicarious learning and social persuasion in group settings may contribute to greater patient activation [47,50]. Thus, the current study utilized tools such as a mobile application (i.e., a WhatsApp chat group) as a supplementary intervention component to encourage patients to share information or advice on how to reduce side effects. Patients with more experience could share their experiences with new patients on addressing the side effects and overcoming the symptoms through self-management. In addition, the present findings suggested that the intervention should be tailored to each patient, which aligned with earlier reports in which tailored interventions helped patients develop specific skills and build confidence [51,52,53,54].

The significant improvement in anxiety and depression levels in this study showed that spending time and frequently interacting with patients positively improved their beliefs regarding managing their illness and reduced their depression scores. Therefore, healthcare providers, such as nurses, can positively influence this population and their treatment outcomes [7,10]. This result was consistent with previous studies conducted by pharmacists in Malaysia [55,56,57] and Western countries [7,10]. Furthermore, the intervention in this study was implemented by an experienced oncology nurse that was caring for cancer patients and understood the challenges patients might face during treatment. In addition, nurses are the most suitable healthcare professionals to execute the self-care education intervention, since they have the most contact with patients.

The study findings suggested that depression and anxiety management required not only the appropriate medication, but also rigorous patient education and counseling [58]. Counseling is an excellent source of mental support, providing patients professional assistance in managing and coping with their situation [59]. In hospitals, healthcare providers play vital roles in collaborating to achieve the best patient outcomes. Nurses can contribute to positive outcomes of chemotherapy by educating, counseling, and motivating cancer patients to comply with their chemotherapy regimens [60]. In this study, the CG had slightly elevated anxiety and depression levels compared to the IG, thus illustrating the central role of anxiety and depression in activation and behavior. Patients who exhibited depressive symptoms were less likely to gain activation and improve their self-management behaviors [14]. Psychological distress screening and treatment are crucial before cancer therapy to ensure a successful intervention and accelerate activation [14]. Therefore, nurses must identify patients who face severe barriers to becoming activated and address the problem before intervention. 

The positive results related to anxiety and depression in the IG may apply to the context of the healthcare system in Malaysia, in which supportive services are limited and emotional support is rarely included in formal cancer care practices [56]. Most cancer care practices in Malaysia, as in other countries, focus on specific symptom burden management and often neglect emotional and social needs due to time constraints and emotionally demanding healthcare tasks. The SCEIP could mediate between treatment-related concerns and the anxiety and depressive symptoms reported by patients with breast cancer [55,56,57]. Furthermore, the study findings confirmed the importance of incorporating emotional support into symptom management interventions to reduce patients’ somatic symptom burdens and psychological distress.

A meta-analysis of randomized controlled trials found that providing both sensory and procedural information reduced anxiety, pain, and distress, in addition to promoting fast recoveries with fewer complications [23]. The review concluded that providing patients with relevant information pertinent to cancer treatment, such as chemotherapy or radiotherapy, was crucial. In addition, patients should be prepared for two critical components (sensory and procedural information), and healthcare providers should address any fears or anxieties before initiating cancer treatment [24]. Only one study agreed with the present conclusion, stating that patients should receive effective pre-treatment education before cancer treatment. In addition, reliable and valid instruments are essential in assessing the preparation for cancer treatment, such as information provision and anxiety management [10]. It is being increasingly recognized that patients require better preparation to deal with their fears and misconceptions regarding the cancer treatment procedure, understand what normal sensations are during and after treatment, and cope with post-treatment side effects [10,24].

In the SCEIP, the CaTS was used to assess patients’ sensory or psychological and procedural concerns, in addition to helping healthcare providers identify gaps in routine preparation for treatment. The tool can identify subgroups of patients with high needs who require more intensive preparation for high-risk and invasive procedures and assist healthcare providers in developing resources and systems to better meet patients’ needs. In addition, studies have shown that patients varied in their preferences for the amount and timing of information [10,24,61,62]. Therefore, the CaTS could be used to assess whether patients require information or support related to their procedure, following the routine preparation for treatment. Furthermore, this information could be useful for nurses to provide tailored strategies or information in preparing patients for their cancer treatments. Ultimately, information curated to patient needs will likely result in improved psychosocial outcomes [10] and better information retrieval [63]. In the present study, both groups showed significant statistical differences in post-test achievement related to the CaTS-SPC and CaTS-PC. Nonetheless, the difference in score increment was higher in the CG than in the IG. This outcome may have been due to the existing practice in the study setting, which allowed patients to discuss and seek any information on chemotherapy side effects and self-care management. 

Finally, several limitations of the study were identified. First, patients were assigned to control and intervention groups based on study stages, thereby limiting obfuscation of the order of patient assignment, which may have denied patients from the control group the opportunity to receive the intervention, and possible biased the results. However, in this study, the inclusion and exclusion criteria were used to ensure sample homogeneity and careful selection of study samples to reduce the possibility of selection bias. Second, this study was conducted at one of the cancer institutes in a public hospital west of Peninsular Malaysia, which may limit generalizability to other parts of the country. Therefore, further studies are proposed to verify the generalizability of the findings by including larger samples. Third, the study did not have a patient-monitoring mechanism to assess patients’ self-care practices at home. For this reason, the present study could not confirm patient compliance with recommended self-care strategies. Furthermore, the study did not collect information regarding the symptom experiences and self-care strategies that the patients adopted at baseline. Similarly, no data were collected when analyzing the proportion of patient engagement and compliance through the WhatsApp conversations in the second phase. Therefore, the researcher could not determine the changes in self-care practices and symptom experiences reported by patients’ post-intervention and whether social media usage influenced their outcomes. Lastly, the stark disparity between the findings of this study and those from Western countries underscored the need to be cautious about the positive results. Thus, the current study needs to be replicated and validated using objective measures of these constructs [64].

## 5. Conclusions

This study provided new insights that filled major gaps in understanding the potential effects of educational and motivational approaches to patient engagement, treatment-related concerns, and psychological distress. Furthermore, the study findings provided preliminary evidence across similar settings and can be applied globally by researchers interested in the potential of the SCEIP in improving patient activation levels, treatment-related concerns, and psychological distress in women with breast cancer undergoing chemotherapy. The involvement of an interdisciplinary team (physicians, psychologists, nurses, physiotherapists, and trainers) who are experts and trained in integrative medicine is crucial for this kind of intervention. Moreover, the positive intervention outcome suggested the need for oncology nurses to act as coordinators in assessing symptoms, symptom burden, and individual patient needs, and in developing plans and support for initial symptom self-care management. Ultimately, this intervention aimed to help patients gain confidence in managing symptoms and engage in effective self-care strategies at home.

## Figures and Tables

**Figure 1 healthcare-10-01572-f001:**
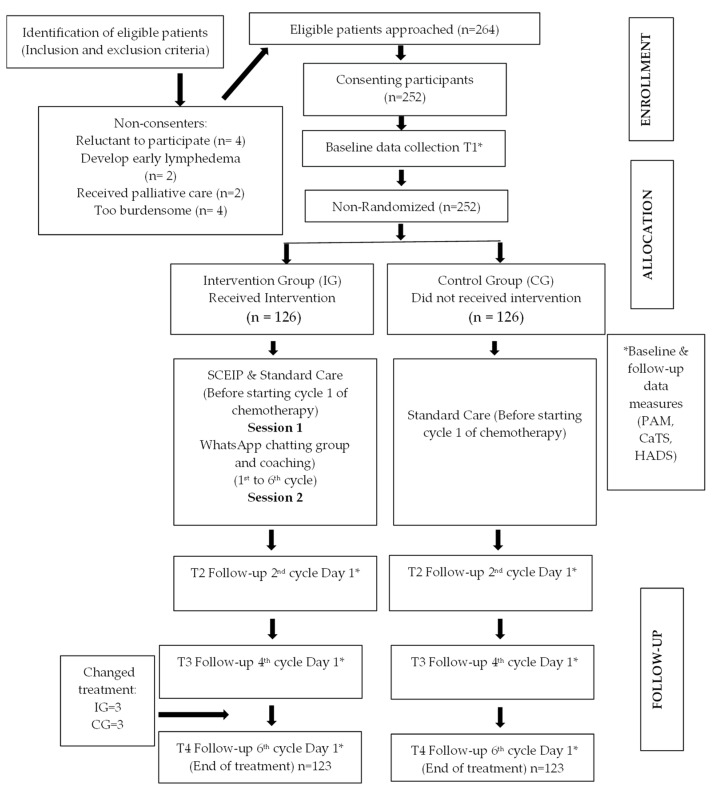
Consort diagram. T, time; Baseline & follow-up data measures, CaTS, Cancer Treatment Survey; HADS, Hospital Anxiety Depression Scale; PAM, Patient Activation Measure.

**Table 1 healthcare-10-01572-t001:** Demographic and medical variables between intervention (IG) and control group (CG).

Variables	IG (*n* = 123)		CG (*n* = 123)		*df*	*t/χ* ^2^	*p-Value*
	*n*	%	*n*	%			
Age, years							
Mean	123	50.14	123	49.89	244	0.192 ª	0.848
SD		±9.48		±11.03			
Age range		30–71		24–75			
Ethnic group							
Malay	80	65	76	61.8	2	0.618 ª	0.734
Chinese	23	18.7	28	22.8			
Indian	20	16.3	19	15.4			
Educational level							
Primary	24	19.5	30	24.4	2	2.008 ª	0.366
Secondary	68	55.3	57	46.3			
Tertiary	31	25.2	36	29.3			
Marital status							
Single/divorced/widowed	26	21.1	25	20.3	1	0.025 ª	0.875
Married	97	78.9	98	79.7			
Employment status							
Working	54	43.9	52	42.3	1	0.066 ª	0.797
Not working	69	56.1	71	57.7			
Income level							
Less than RM 1500	31	25.2	35	28.5	2	2.492 ª	0.288
RM 1501–3000	37	30.1	45	36.5			
More than RM 3001	55	44.7	43	35.0			
Menopausal status							
Pre-menopausal	65	52.8	65	52.8	1	0.000 ª	1.000
Post-menopausal	58	47.2	58	47.2			
ECOG performance status							
0	53	43.1	57	46.3	1	0.263 ª	0.608
1	70	56.9	66	53.7			
Staging of cancer							
I	6	4.9	3	2.5	2	1.011 ^b^	0.677
II	47	38.2	48	39.0			
III	70	56.9	72	58.5			
Chemotherapy regimen							
Anthracycline alone	60	48.8	73	59.3	1	2.766 ª	0.125
Anthracycline- and taxane-based	63	51.2	50	41.7			

Note: ª independent *t*-test; ^b^ Fisher’s exact test; SD = standard deviation; ECOG = Eastern Cooperative Oncology Group.

**Table 2 healthcare-10-01572-t002:** Effect of SCEIP on PAM between and within intervention and control groups across time.

Measure	Time	Group	
		Intervention M (±SD)	Control M (±SD)
Patient Activation Level	Baseline (T1)	64.79 (7.66) ^a,x^	64.92 (8.63) ^a,x^
	Cycle 2 (T2)	66.97 (7.87) ^a,x,y^	62.69 (6.80) ^b,x,y^
	Cycle 4 (T3)	67.00 (7.93) ^a,y^	62.34 (8.26) ^b,y^
	Cycle 6 (T4)	68.13 (7.84) ^a,z^	61.23 (2.54) ^b,z^

Note: Means with different letters were statistically significant at *p* < 0.05 using the Bonferroni test; ^a,b^: between-group comparison; ^x,y,z^: within-group comparison.

**Table 3 healthcare-10-01572-t003:** Effect of SCEIP on HADS between and within intervention and control groups across time.

Measure	Time	Group	
		Intervention M (±SD)	Control M (±SD)
Anxiety	Baseline (T1)	5.84 (3.54) ^a,x^	5.98 (3.63) ^a,x^
Depression		4.10 (3.08) ^a,x^	4.60 (3.30) ^a,x^
Anxiety	Cycle 2 (T2)	2.80 (2.19) ^a,x,y^	4.72 (3.05) ^b,x,y^
Depression		3.27 (2.51) ^a,x,y^	4.77 (3.42) ^b,x,y^
Anxiety	Cycle 4 (T3)	4.24 (3.09) ^a,y^	5.01 (3.40) ^b,y^
Depression		3.76 (2.52) ^a,y^	4.55 (3.40) ^b,y^
Anxiety	Cycle 6 (T4)	3.98 (2.52) ^a,z^	5.13 (3.30) ^b,z^
Depression		4.03 (2.73) ^a,z^	4.66 (3.28) ^b,z^

Note: Means with different letters were statistically significant at *p* < 0.05 using the Bonferroni test; ^a,b^: between-group comparison; ^x,y,z^: within-group comparison.

**Table 4 healthcare-10-01572-t004:** Effect of SCEIP on CaTS between and within intervention and control groups across time.

Measure	Group	Pre-Test (T1) M (±SD)	*p-Value* _(Between Group)_	Post-Test (T4) M (±SD)	*p-Value* _(Between Group)_	*p-Value* _(Within Group)_
CaTS-SPC	Control	3.73 (0.38)	0.955	3.14 (0.82)	0.042 *	<0.001 **
	Intervention	3.72 (0.42)		2.90 (1.04)		<0.001 **
CaTS-PC	Control	4.18 (0.41)	0.250	3.83 (0.23)	<0.001 **	<0.001 **
	Intervention	4.24 (0.42)		3.21 (1.27)		<0.001 **

* Significant difference at *p* < 0.05, ** significant difference at *p* < 0.001.

## Data Availability

Data are available upon request from the authors.

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
