# Peer review of "Effects of Self-Care Education Intervention Program (SCEIP) on Activation Level, Psychological Distress, and Treatment-Related Information"

_healthcare, 2022, doi:10.3390/healthcare10081572_

Round 1
Reviewer 1 Report
The study entitled “ Effects of Self-Care Education Intervention Program (SCEIP) on Activation Level, Psychological Distress and Treatment-Related Information“ offers an interesting and worthwhile contribution to the extant literature. Its reliance on an understudied patient population is one of its most meaningful contributions. There are, however, a few aspects of the manuscript that demand clarification.
The meaning of the term“patient activation” needs to be clarified. Do the authors mean “patients’ activity level”?
2 A clear description of the main properties of the Self-Care Education Intervention Program (SCEIP) that serves as the independent variable is to be provided in the introduction.
3 The selection of variables upon which the treatment effect is measured may benefit from being entered into a conceptual framework, such as a model/theory of a person’s wellbeing. Otherwise, it is not entirely clear why the authors selected some factors (i.e., patients’ activity levels, psychological distress, and treatment-related concerns in women with breast cancer), and not others.
4 The authors state that “this study did not include randomization in the intervention because of concerns about contamination between the intervention and control arm which might have arisen in the case of a randomized controlled trial”. The authors need to provide additional information that justifies their decision not to rely on random assignment at the start of the study.
5. Social desirability effects need to be discussed in reference to the uncovered differences between control and treatment conditions. Social desirability is a particularly noteworthy concern since the authors did not rely on random assignment.
6. The section devoted to conclusions is excessively brief and general. Are there recommendations that can be made based on the study’s results? Can a person-oriented approach, instead of a variable-oriented approach provide additional information regarding participants’ unique responses to the selected intervention?
7. Thorough proofreading of the manuscript is advised. Word choices may need to be carefully checked.
Author Response
Good Day Professor/Doctor,
Thank you for your comments and important points highlighted in the manuscript. The response of the comment given is at the attached document. Thank you again.

Reviewer 2 Report
An excellent intervention well described. There are a number of minor points. (1) It is not clear to me how patients were allocated between treatment and control. Was it sequential time based cohorts with a gap? was the consent the same for both groups?
(2) What does sensory in sensory and and procedural information mean.
(3) The Bonferroni method correction mentioned in the results is not mentioned in the methods.
(4) I could not see any information on compliance or engagement. What proportion of patients engaged in the whats app conversations in the second phase - what measures of participation are there.
(5) Completeness of data and handling of missing data. I couldn't see in the methods how missing data was handled and i couldn't see in the results what numbers of patients filled in the assessment at the latter time points. This is key information in a longitudinal design.
(6) Context of data collection - how did the participants fill out the questionairres - was there potential for influence of the responses by the investigators - what protection was there aganist this.
(7) intervention delivery - who delivered the intervention - was it one person ? a team
(8) funding who paid for this - is there any potential for bias in reporting
(9) Time cost - how long did it take to fill out the questionairres - how much support was needed.
(10) the justification of the design avoiding randomisation is inadequate and the discussion of the way in which the unrandomised design could have influenced the results is inadequate. It appears that one researcher collected subjective data from patients. There are multiple ways this could have influenced results without further protections. Consequently the conclusions need to be toned down.
(11) For a complex intervention the discussion needs to acknowledge uncertainty as to what parts of the intervention were responsible for any possible effect.
Author Response
Dear Professor/Doctor,
Thank you for your comments and important points highlighted. The response of the comment is at the attached document. Thank you again.
